# Living Chain-Walking (Co)Polymerization of Propylene and 1-Decene by Nickel *α*-Diimine Catalysts

**DOI:** 10.3390/polym12091988

**Published:** 2020-08-31

**Authors:** Pei Li, Xiaotian Li, Shabnam Behzadi, Mengli Xu, Fan Yu, Guoyong Xu, Fuzhou Wang

**Affiliations:** 1Institutes of Physical Science and Information Technology, Anhui University, Hefei 230601, China; lipei12077201@163.com (P.L.); lxt930415@163.com (X.L.); xumengli@126.com (M.X.); yufan0124@126.com (F.Y.); 2CAS Key Laboratory of Soft Matter Chemistry, Hefei National Laboratory for Physical Sciences at the Microscale, Department of Polymer Science and Engineering, University of Science and Technology of China, Hefei 230026, China; Behzadi@mail.ustc.edu.cn

**Keywords:** living (co)polymerization, chain-walking, propylene, 1-alkene, branched polyolefins

## Abstract

Homo- and copolymers of propylene and 1-decene were synthesized by controlled chain-walking (co)polymerization using phenyl substituted α-diimine nickel complexes activated with modified methylaluminoxane (MMAO). This catalytic system was found to polymerize propylene in a living fashion to furnish high molecular weight ethylene-propylene (EP) copolymers. The copolymerizations proceeded to give high molecular weight P/1-decene copolymers with narrow molecular weight distribution (*M*_w_/*M*_n_ ≈ 1.2), which indicated a living nature of copolymerization at room temperature. The random copolymerization results indicated the possibility of precise branched structure control, depending on the polymerization temperature and time.

## 1. Introduction

Cationic late-transition metal catalyzed homo- and copolymerization of alkenes such as ethylene and propylene have received substantial attention in polymer science and materials chemistry [1,2,3,4,5]. Specifically, polypropylene (PP) is a very popular plastic material used for many applications, namely, in electrical devices, automotive parts, food packaging, household equipment, and many others [6]. At present, due to the epidemic situation of novel coronavirus pneumonia in the world, epidemic prevention materials in many countries are in short supply, especially for medical grade polypropylene raw materials for the production of medical masks.

In the field of polyolefin research [7,8,9,10,11,12,13,14,15,16,17,18,19,20], many studies have reported on the homopolymerization of ethylene [21,22,23,24], different higher 1-alkene [25,26,27,28,29,30,31], and linear internal alkenes [32,33,34] using Brookhart-type α-diimine nickel and palladium catalysts [35,36,37,38,39,40,41] because of the chain-walking process [42,43,44,45]. The interest in highly branched polyolefins such as dendrimer and hyper branched polymers has increased obviously in recent years owing to these unique physical properties, chemical properties, and their potential applications as adhesives, lubricants, and paints [7]. Compared with a commercial linear polyethylene (LPE), these branched polyolefin-based materials with a chain topology possess unique physical properties stemming from their architectures [2,7], and provide the potential to increase polymeric melt fracture resistance, elasticity, paintability, and compatibility with other materials. However, propylene polymerization by imine-based nickel catalysts has been reported only in a limited number of cases [46,47,48], and no example for propylene copolymerization with higher 1-alkene. Polymerization of propylene with α-diimine nickel complexes produced PPs with different structures depending on the *ortho*-substituent groups of the catalysts and the polymerization temperature [44]. Suitable catalysts polymerize propylene to a syndiotactic polymer at a low temperature [46] but to regioregular polymers at higher temperature.

Chain-walking alkene polymerization [44] can give polymers unique structures, which cannot be obtained by common vinyl polymerization. Chain-branching formation in the polymerization of linear 1-alkene using cationic nickel α-diimine catalysts is shown in Scheme 1 [49]. This mechanism involves 1,2- and 2,1-insertion followed by chain-walking behavior [44], in which the active metal undergoes chain-walking to the terminal or internal carbon followed by monomer insertion [49]. A 2,1-insertion followed by complete chain-walking installs a long methylene sequence [44], and 1,2-insertion gives the methyl branch in 2,*ω*-enchainment and the *n*-alkyl branch without chain-walking behavior. While some methyl and alkyl branches are also derived from a small amount of the partial chain-walking to the 1-alkene internal carbon [48], the product properties strongly depend on the microstructure of different branched polyolefins [1]. Generally, the highly branched polyolefins containing methyl and alkyl branches are amorphous, and the chain-straightened linear polymeric products are semicrystalline. For instance, Coates et al. demonstrated that 1-butene polymerization via stereoretentive chain-walking using cationic nickel *α*-diimine catalysts to produce semicrystalline isotactic 4,2-poly(1-butene) [50]. Cationic “sandwich” *α*-diimine nickel conducted accurate chain-walking polymerization of higher 1-alkenes to generate semicrystalline “polyethylene” [51].

Our group reported recently that the introduction of phenyl groups in *α*-diimine nickel system can efficiently improve catalytic performances with a fast chain-walking process [52,53,54,55]. Consequently, these catalysts conducted polymerizations of ethylene [52,53], 4-methyl-1-pentene [54] and linear 1-alkenes [55] to gain high molecular weight polymers and proved living behavior at low temperature. In this work, therefore, the propylene homopolymerization and copolymerization with 1-decene were conducted by phenyl substituted nickel *α*-diimine complexes activated by MMAO.

## 2. Experimental Section

### 2.1. General Considerations

All manipulations, unless otherwise mentioned, were carried out using standard Schlenk or in glovebox. Nuclear magnetic resonance (^1^H, ^13^C NMR) spectra were recorded on a Varian 400 NMR instrument (Varian, Inc., Palo Alto, CA) at room temperature and 50 °C, respectively, using CDCl_3_ as a solvent and tetramethylsilane (TMS) as internal standard for the compounds. Gel permeation chromatography (GPC) analyses of the molecular weight and molecular weight distribution of the polymers were performed on a Tosoh HLC-8320GPC chromatograph (Tosoh Asia Pte. Ltd., Tokyo, Japan) at 40 °C using THF as an eluent. Differential scanning calorimetry (DSC) was performed by a SII EXSTAR6000 system (Hitachi, Tokyo, Japan). Research-grade propylene was purified by passing it through dehydration column of ZHD-20 and deoxidation column of ZHD-20A. MMAO (6.5 wt % Al, 2.17 M in toluene) was donated by Tosoh-Finechem Co (Tokyo, Japan). Toluene was dried with sodium/benzophenone under nitrogen atmosphere and distilled before use. *n*-Hexane, CDCl_3_ and CH_2_Cl_2_ were purified over 4 Å molecular sieves. 1-Decene was purchased from Kanto Chemical Co on Aldrich Chemical Company (Tokyo, Japan) were dried over CaH_2_ and distilled before use. Nickel complexes **1**–**5** were synthesized according to the literature [55]. Other chemicals were commercially obtained and purified with common procedures.

### 2.2. Polymerization Procedure

The propylene polymerization/copolymerization experiments were performed in a 100-mL glass reactor equipped with a temperature controller and a magnetic stirrer. A 30 mL amount of toluene was added to the dried reactor reactor under *N*_2_ atmosphere.

Propylene polymerization: Propylene (1.2 bar) was introduced to the reactor kept at polymerization temperature after the toluene was saturated with propylene; MMAO was added and the solution was stirred for 10 min. Then, the catalyst solution (10 μmol) in toluene was injected into the polymerization system with a syringe to start polymerization. After the desired amount of polymerization time, the pressure vessel was vented and the reaction was quenched by addition of 5% acidic methanol (HCl/MeOH).

Propylene/1-decene copolymerization: 1-Decene was added to the toluene via a syringe, and the resulting solution was saturated with propylene (1.2 bar) under vigorous stirring for 10 min. Then MMAO was added and the solution was stirred for 10 min. Then, the catalyst solution (10 μmol) in toluene was injected into the polymerization system to start polymerization. After the desired amount of polymerization time, the pressure vessel was vented and the reaction was quenched by addition of 5% acidic methanol (HCl/MeOH).

The PPs and copolymers obtained were filtered from solution, washed with methanol, and dried in a vacuum oven to constant weight.

### 2.3. Time-Course of Propylene and 1-Decene Copolymerization

Copolymerization of propylene and 1-decene was performed in a 100-mL glass reactor equipped with a magnetic stirrer. A 30 mL amount of toluene was added to the fully dried reactor under *N*_2_ atmosphere. After the reactor was set at polymerization temperature, 1-decene (12 mmol) was added to the toluene via a syringe, and the resulting solution was saturated with propylene (1.2 bar) under vigorous stirring for 10 min. Then MMAO was added and the mixture was stirred for 10 min. The catalyst **2** (12 μmol) solution in toluene was added to the reactor (total volume = 40 mL) to start polymerization. The reaction solution (6.6 mL) was sampled six times via a syringe at different polymerization time (5 min, 10 min, 30 min, 50 min, 70 min and 90 min), and the reaction mixture was terminated with 40 mL of a 3% HCl/MeOH solution. The copolymers obtained were filtered from solution, washed with methanol, and dried in a vacuum oven to constant weight.

## 3. Results and Discussion

### 3.1. Synthesis and Characterization of the Nickel Complexes

Our previous results have shown that fast chain-walking process can be efficiently modulated using phenyl substituted Brookhart-type nickel *α*-diimine catalysts [52,53,54,55]. Complex **2** activated with MMAO exhibited high activity for ethylene, linear, and branched 1-alkenes polymerizations in a living manner at room temperature. Four phenyl substituted *α*-diimine complexes were therefore synthesized as a precursor for propylene polymerization with modified MMAO, and these nickel complexes **1**–**4** [55] are summarized in Scheme 2. The corresponding nickel catalyst {[(2,4,6-Me_3_C_6_H_2_N=C)_2_Nap]NiBr_2_ (**5**)} [56] was also synthesized for comparison.

Single-crystal of **3** was obtained by slowly diffusing of *n*-hexane into the CH_2_Cl_2_ solution of complex at room temperature, and its molecular structure was confirmed by X-ray diffraction (Figure 1). Crystal data, data collection, and refinement parameters are listed in Appendix A (see the Appendix A). The geometry at the Ni center is pseudo-tetrahedral, showing pseudo-*C*_2_-symmetry. The bond length of N1−C20 [1.298(8) Å] and N2−C31 [1.264(8) Å] have typical imine double bonds character, respectively. In the solid state, the Ni1−N1 and Ni1−N2 bond distances of 2.036(6) and 2.017(6) Å in **3** are slightly narrower than the value of 2.041(5) Å for analogue **2 [56]**. In addition, the N1–Ni1–Br3 angles of 117.99(7)° for **3** is also approximate to those for complex **2** (112.96°). The conjugation effect of phenyl substituents in the *para*- and *ortho*-aryl position of *α*-diimine ligand can be clearly observed in these molecular structures, expected to improve the catalyst performance for propylene (co)polymerization.

### 3.2. Catalytic Polymerization of Propylene

Upon activation with MMAO, polymerization of propylene was examined by complexes **1**–**5** with the [Al]/[Ni] ratio of 300 at 1.2 bar of propene for 30 min, and the results are listed in Table 1. The effect of reaction temperature was examined by varying the temperature from 0 to 50 °C (Table 1, entries 1–3). Both the highest activity and the highest molecular weight of 13.4 × 10^4^ g mol^−1^ were observed with **2**-MMAO at 25 °C (Table 1, entry 2). The narrowest molecular weight distribution of 1.06 was observed at 0 °C (Table 1, entry 1). Increasing the polymerization temperature further to 50 °C (Table 1, entry 3), the molecular weight decreased dramatically, which was accompanied by a slight broadening of the *M*_w_/*M*_n_ value.

The effect of different catalyst precursors **1***–***5** was also studied at 25 °C (Table 1, entries 2 and 4–7). Phenyl substituted **1**–**4** produced the PPs in moderate yields with high molecular weights of (11.9–19.1) × 10^4^ g mol^−1^ and narrow molecular weight distribution (*M*_w_/*M*_n_ = 1.16–1.48, Table 1, entries 2 and 4–6). The catalytic activities decreased in the order, **2** ≥ **1** ≥ **3** ≥ **4**, and the *M*_n_ values increased with increasing the ligand bulkiness (**4** ≥ **3** ≥ **2** ≥ **1**). Complex **2** produced higher molecular weight PP in highest yield than those of one methyl or phenyl groups (**1**, **3** and **4**, Table 1, entries 4–6). The highest molecular weight PP obtained by **4** (Table 1, entry 6), indicating the rate of chain propagation was greatly promoted by the bulky *ortho*-phenyl substituent on the N-aryl moiety, which should retard chain-transfer reactions [44]. Complex **2** with the conjugation ability of the *para*-phenyl substituent exhibited higher activity than the corresponding methyl-substituted **5** (Table 1, entries 2 vs. 7), as already observed in polymerization of ethylene and higher 1-alkenes [55]. The higher molecular weight PP with narrower molecular weight distribution was obtained by **2** suggests that phenyl substituent should suppress chain-transfer reactions [44].

The effect of polymerization time in propylene polymerization was also studied by **2**-MMAO at 25 °C (Table 1, entries 2 and 8–11). The yields and the *M*_n_ value increased with the prolongation of polymerization time. Figure 2a shows the plot of *M*_n_ increased linearly polymerization time accompanied with keeping narrow *M*_w_/*M*_n_ = 1.10–1.25 (Appendix A from Appendix A), and the *N* value was almost constant. This polymerization result verified that the propylene polymerization by **2**-MMAO proceeded in a living manner at room temperature within a certain period of time.

^1^H NMR spectroscopy analyses [57,58] showed that the obtained PPs proceeded only methyl group, while the total branching densities (245–267/1000 C, Table 2) were less than the expected value (333/1000 C), indicating the 1,3-enchainment of propylene via 2,1-insertion followed by chain-straightening. The 1,3-enchainment increased slightly with increasing reaction time and temperature (Table 2). Upon comparison with common polypropylene (PP, *T*_g_ = −10~−5 °C), the produced PPs had low *T*_g_ of approximately −29.4~−24.1 °C and no *T*_m_ (Table 2), indicating that the PPs were amorphous.

The microstructure analyses of the polypropylenes were further studied by ^13^C NMR spectroscopy [59,60]. ^13^C NMR spectra of polypropylenes obtained by **2**-MMAO at 0 and 25 °C are shown in Figure 3 (Table 1, entries 1 and 2), where only normal methyl branch was observed at 0 °C (Figure 3a), while 2-methylhexyl structure was observed at room temperature (Figure 3b). When nickel metal migrates forward along the newly 2,1-added propylene unit, ethylene-type unit coming from 1,3-enchainment can be produced, and backward along the polymer chain, longer branched chains can be formed [46]. Consequently, the microstructures of the polypropylenes were similar to the ethylene-propylene (EP) copolymers. Therefore, **2**-MMAO was found to polymerize propylene in a living fashion to furnish EP copolymers.

### 3.3. Copolymerization of Propylene and 1-Decene

Copolymerization of propylene and 1-decene (D) was catalyzed by **2**-MMAO at 0 and 25 °C with the [Al]/[Ni] ratio of 300, and the results are listed in Table 3. In comparison with the 1-decene homopolymerization under the same conditions [55], P/1-decene copolymerization exhibited higher activity and produced copolymers with higher molecular weight and narrow *M*_w_/*M*_n_ values. In the P/1-alkene copolymerization study, similar trends were observed to those for propylene and 1-decene homopolymerization. For instance, **2**-MMAO exhibited higher activity and higher molecular weight than the corresponding methyl-substituted **5** (Table 3, entries 1 vs. 3).

The P/1-decene copolymer obtained with **2**-MMAO at 25 °C showed narrow molecular weight distribution of 1.14 (Table 3, entry 1). Time-course of polymerization was investigated with **2**-MMAO at 25 °C under the same conditions by sampling method, and the results are summarized in Table 4. The plot of *M*_n_ against the P-D copolymer yield shows a good linear relationship accompanied by a very narrow polydispersity (*M*_w_/*M*_n_ = 1.10–1.12, Figure 2b), keeping the number of polymer chains (*N*) constant. This result verified that the P/1-decene copolymerization with **2**-MMAO proceeded in a living/controlled manner at 25 °C within a certain period of time. To the best of our knowledge, this is the first example of late transition metal catalyzed living copolymerization of propylene and 1-decene at room temperature.

The copolymerization results indicated the possibility of precise microstructure control, depending on the polymerization temperature, which in turn strongly affects the physical polymer properties. The branching densities of the obtained P/1-decene copolymers thus obtained are shown in Table 3. P/1-decene copolymers possessed highly B values in comparison with the corresponding poly(1-decene)s [55]. The branching densities of the P/1-decene copolymer obtained at 0 °C (187/1000 C, Table 3, entry 2) was much higher than those of at room temperature (169/1000 C, Table 3, entry 1). Therefore, the branching density can be controlled by the polymerization temperature. The produced copolymers had *T*_g_ of approximately −49 °C no *T*_m_ (Table 3), indicating that the P/1-decene copolymers were amorphous.

The methyl branches were formed by predominant P/1-alkene 1,2-insertion and precise chain-walking to form a *ω*,2-enchaninment (Scheme 1), while the long branches were formed by 1,2-insertion of higher 1-decene. P/1-decene copolymers have almost exclusively methyl and longer branches (Figure 4), because the 1-decene give octyl (C_8_) branch separately in 1,2-insertion. For example, the ^13^C NMR spectra of the copolymers obtained by **2**-MMAO, and showed that the poly(P-*co*-D) produced at 25 °C possessed 170 branches/1000 carbons including 133 methyl and 37 longer chains (C_8_) (Figure 4), whereas poly(1-decene) possessed 83 branches/1000 carbons including 40 methyl, 4 propyl, and 39 longer chains. There were no carbon atoms of adjacent methyl branches (14.5–17.5 ppm) and ethyl branch in the ^13^C NMR spectra. Therefore, the 2,1-insertion of propylene and 1-decene always evolves into a 1,3- and 1,8-enchainment to give methylene sequences (Scheme 1). The branch-type distribution depended on the reaction temperature, and poly(P-*co*-D) produced at 0 °C possessed 186 branches/1000 carbons including 146 methyl and 40 longer chains (Figure 4b).

## 4. Conclusions

In summary, we conducted the living chain-walking polymerization of propylene using phenyl substituted *α*-diimine nickel catalysts in combination with MMAO to generate high molecular weights amorphous “ethylene-propylene (EP) copolymers” at room temperature. Copolymerization of propylene and 1-decene was successfully achieved using **2**-MMAO, produced highly branched P/1-decene copolymers with high molecular weight and narrow molecular weight distribution. At room temperature, living copolymerization of propylene and 1-decene was observed. The ^13^C NMR spectra of the P/1-decene copolymers obtained showed a microstructure almost exclusively composed of methyl and octyl (C_8_) branches.

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
