# Peer review of "Living Chain-Walking (Co)Polymerization of Propylene and 1-Decene by Nickel α-Diimine Catalysts"

_polymers, 2020, doi:10.3390/polym12091988_

Round 1
Reviewer 1 Report
This is a well-prepared article describing the copolimerazion of propylene and 1-decene in presence of nickel diamine catalysts. The objectives of the article may be better specified, in particular:
the authors should explain in the introduction the reasons to perform the co-polymerization reactions using 1-decene. Has this terminal olefin been chosen for a specific reason in terms of expected polymer properties? Which benefits are expected with respect to the copolymerization of 4-methyl-1-pentene already described (ref. 54).
Experimental section, first paragraph: NMR spectra cannot be performed in CDCl3 at 80 °C. Please explain.
Minor typographical and language changes are as follows:
Abstract, line 3: This catalytic…..
Introduction line 3:……applied…..used…
Line 5: …..Nnovel Coronavirus……(no reason to use capital letter for novel)
…..substituent groups………
Page 2, line 8:….strongly depend………
Line 9: there is no verb in the sentence.
Page 2 of introduction, last sentence: …can efficiently improve the catalytic performances…..
Experimental:
Page 3, line 4: … a deoxygenated and a dry column..... containing what?
What is ….acidic methanol (5%)?
The conjugation effect OF phenyl substituentS in the para and ortho-aryl position of α-diimine ligand can be clearly observed IN these molecular structures, expectED to improve the catalyst performance for propylene (co)polymerization.
Page 5, line 6: ……..should retard chain……
Line 10: ……..weight distribution was obtained by 2 suggestS that phenyl…..
Page 6, line 3:……less methyl group…??
Line 4: Upon comparison with common polypropylene…
Second paragraph: …..more long branch…..longer branched chains…..
Page 7:…..Time-course of polymerization….
Page 8, two lines after figure: -enchainment
For example, the 13C NMR spectra of the copolymers obtained by 2-MMAO, and showed THAT the poly….
……1,8-enchainment…..
Author Response
Title: “Living Chain-Walking (Co)Polymerization of Propylene and 1-Decene by Nickel α-Diimine Catalysts”
Polymers 894363
Dear Reviewers,
Thanks for your useful comments. We have revised the manuscript accordingly, and detailed corrections are listed below point by point:
Reviewer: 1
This is a well-prepared article describing the copolimerazion of propylene and 1-decene in presence of nickel diamine catalysts. The objectives of the article may be better specified, in particular: the authors should explain in the introduction the reasons to perform the co-polymerization reactions using 1-decene.
Has this terminal olefin been chosen for a specific reason in terms of expected polymer properties? Which benefits are expected with respect to the copolymerization of 4-methyl-1-pentene already described (ref. 54).
Thanks for your useful suggestion. The introduction section has been revised according to your suggestions:
The interest in highly branched polyolefins such as dendrimer and hyper branched polymers increased obviously in recent years owing to these unique physical properties, chemical properties and potential applications as adhesives, lubricants, and paints [7]. Compared with a commercial linear polyethylene (LPE), these branched polyolefin-based materials with a chain topology possess unique physical properties stemming from their architectures [2,7], and provides the potential to increase polymeric melt fracture resistance, elasticity, paintability, and compatibility with other materials. However, propylene polymerization by imine-based nickel catalysts has been reported only in a limited number of cases [46–48], and no example for propylene copolymerization with higher 1-alkene.
Experimental section, first paragraph: NMR spectra cannot be performed in CDCl3 at 80 °C. Please explain.
Thanks a lot for catching this. The experimental section has been revised.
"80 °C" has been corrected to "50 °C".
1H and 13C NMR spectra of compounds and polymers were recorded on a Varian 400 NMR instrument at room temperature and 50 °C, respectively, using CDCl3 as a solvent and tetramethylsilane (TMS) as a reference for the compounds.
Minor typographical and language changes are as follows:
Thanks a lot for catching these typographical and language errors.
Abstract, line 3: This catalytic…..
"These catalytic system" has been corrected to "This catalytic system".
Introduction line 3:……applied…..used…
"applied" has been corrected to "used".
Line 5:…..Nnovel Coronavirus……(no reason to use capital letter for novel)
"Nnovel Coronavirus Pneumonia" has been corrected to "novel coronavirus pneumonia".
…..substituent groups
"ortho-substituented groups" has been corrected to "ortho-substituent groups".
Page 2, line 8:….strongly depend………
"strongly depends" has been corrected to " strongly depend".
Line 9: there is no verb in the sentence.
Generally, the highly branched polyolefins containing methyl and alkyl branches are amorphous, and the chain-straightened linear polymeric products are semicrystalline.
Page 2 of introduction, last sentence: …can efficiently improve the catalytic performances…..
"can be efficiently improved the catalytic performances" has been corrected to "can efficiently improve the catalytic performances".
Experimental:
Page 3, line 4:… a deoxygenated and a dry column..... containing what?
Research grade propylene was purified by passing it through dehydration column of ZHD-20 and deoxidation column of ZHD-20A.
What is ….acidic methanol (5%)?
5%.acidic methanol (HCl/MeOH)
The conjugation effect OF phenyl substituentS in the para and ortho-aryl position of α-diimine ligand can be clearly observed IN these molecular structures, expectED to improve the catalyst performance for propylene (co)polymerization.
This sentence has been revised according to your suggestion:
The conjugation effect of phenyl substituents in the para- and ortho-aryl position of α-diimine ligand can be clearly observed in these molecular structures, expected to improve the catalyst performance for propylene (co)polymerization.
Page 5, line 6: ……..should retard chain……
"should retarded chain" has been corrected to "should retard chain".
Line 10: ……..weight distribution was obtained by 2 suggestS that phenyl…..
"suggest" has been corrected to "suggests".
Page 6, line 3:……less methyl group…??
…the obtained PPs proceeded only methyl group, while the total branching densities (245−267/1000 C, Table 2) were less than the expected value (333/1000 C).
Line 4: Upon comparison with common polypropylene…
This sentence has been revised according to your suggestion.
Second paragraph: …..more long branch…..longer branched chains…..
"more long branch" has been corrected to "longer branched chains".
Page 7:…..Time-course of polymerization….
"couse" has been corrected to "course".
Page 8, two lines after figure: -enchainment
For example, the 13C NMR spectra of the copolymers obtained by 2-MMAO, and showed THAT the poly….
……1,8-enchainment…..
"1,8-enchainement " has been corrected to "1,8-enchainment".
Reviewer: 2
The authors make a rather routine contribution without reporting new aspects. The catalyst precursors and the cocatalyst are known. The method is known. The only new aspect is the application of 1-decene as comonomer in combination with propene. The quality of the presentation can be improved (proper use of articles, singular and plural. The references are not balanced: 75% have Chinese authors (44 out of 58) and the impression could be that this chemistry was created by the mentioned research groups. Further comments:
Thank you for your useful comments. We have revised this manuscript. Propylene polymerization by chain-walking Ni catalysts has been reported only in a limited number of cases [46–48], and no example for propylene copolymerization with higher 1-alkene.
We have added some important literature of chain-walking propylene polymerization in the reference section.
See Reference:
- Jeon, M.; Han, C.J.; Kim, S.Y. Polymerizations of propylene with unsymmetrical (α-diimine)nickel(2) catalysts. Res. 2006, 14, 306−311.
- Pellecchia, C.; Zambelli, A. Syndiotactic-specific polymerization of propene with a Ni-based catalyst. Rapid Commun. 1996, 17, 333–338.
- Pellecchia, C.; Zambelli, A.; Oliva, L.; Pappalardo, D. Syndiotactic-specific polymerization of propene with nickel-based catalysts. 2. Regiochemistry and stereochemistry of the initiation steps. Macromolecules 1996, 29, 6990–6993.
-The polymerization of propene in toluene is unusual because of the potential influence of the solvent. Most of the reports are dealing with liquid propene and a polymerization period of 60 minutes
Thanks for your useful suggestion.
Before propylene polymerization: Propylene was introduced to the reactor kept at polymerization temperature after the toluene was saturated with propylene, MMAO was added and the solution was stirred for 10 min. In toluene, propylene is already in a saturated state.
In addition, our previous research results have shown that it is very easy to catalyze the polymerization of 1-olefins by phenyl substituted nickel α-diimine catalysts, so the P/D copolymerization can be successfully completed within 30 minutes at room temperature.
- Scheme 2: For R exponents should be used and no indices
Scheme 2. Chain-walking polymerization of propylene catalyzed with α-diimine nickel catalysts 1–5 [55,56].
- P. 3: The detailed description of the polymerization/copolymerization reactions can be summarized in order to avoid repetitions
Thanks for your useful suggestion. Propylene polymerization/copolymerization reactions have revised.
- P. 2: l.12 from the bottom: homo polymerization
Thanks a lot for catching this.
"home-polymerization" has been corrected to "homo-polymerization".
- P. 7, Table 3, top line: Branches has the exponent "d"
"c" has been corrected to "d".
- P. 8, l.5: enchainment
"1,8-enchainement " has been corrected to "1,8-enchainment".
The revised manuscript has been resubmitted to ‘polymers’. We look forward to your positive response.
Best wishes,
Institute of Physical Science and Information Technology,
Anhui University, Hefei 230601, Anhui, China.

Reviewer 2 Report
The authors make a rather routine contribution without reporting new aspects. The catalyst precursors and the cocatalyst are known. The method is known. The only new aspect is the application of 1-decene as comonomer in combination with propene. The quality of the presentation can be improved (proper use of articles, singular and plural. The references are not balanced: 75% have Chinese authors (44 out of 58) and the impression could be that this chemistry was created by the mentioned research groups. Further comments:
- The polymerization of propene in toluene is unusual because of the potential influence of the solvent. Most of the reports are dealing with liquid propene and a polymerization period of 60 minutes
- Scheme 2: For R exponents should be used and no indices
- P. 3: The detailed description of the polymerization/copolymerization reactions can be summarized in order to avoid repetitions
- P. 2: l.12 from the bottom: homo polymerization
- P. 7, Table 3, top line: Branches has the exponent "d"
- P. 8, l.5: enchainment
Author Response
Title: “Living Chain-Walking (Co)Polymerization of Propylene and 1-Decene by Nickel α-Diimine Catalysts”
Polymers 894363
Dear Reviewers,
Thanks for your useful comments. We have revised the manuscript accordingly, and detailed corrections are listed below point by point:
Reviewer: 2
The authors make a rather routine contribution without reporting new aspects. The catalyst precursors and the cocatalyst are known. The method is known. The only new aspect is the application of 1-decene as comonomer in combination with propene. The quality of the presentation can be improved (proper use of articles, singular and plural. The references are not balanced: 75% have Chinese authors (44 out of 58) and the impression could be that this chemistry was created by the mentioned research groups. Further comments:
Thank you for your useful comments. We have revised this manuscript. Propylene polymerization by chain-walking Ni catalysts has been reported only in a limited number of cases [46–48], and no example for propylene copolymerization with higher 1-alkene.
We have added some important literature of chain-walking propylene polymerization in the reference section.
See Reference:
- Jeon, M.; Han, C.J.; Kim, S.Y. Polymerizations of propylene with unsymmetrical (α-diimine)nickel(2) catalysts. Res. 2006, 14, 306−311.
- Pellecchia, C.; Zambelli, A. Syndiotactic-specific polymerization of propene with a Ni-based catalyst. Rapid Commun. 1996, 17, 333–338.
- Pellecchia, C.; Zambelli, A.; Oliva, L.; Pappalardo, D. Syndiotactic-specific polymerization of propene with nickel-based catalysts. 2. Regiochemistry and stereochemistry of the initiation steps. Macromolecules 1996, 29, 6990–6993.
-The polymerization of propene in toluene is unusual because of the potential influence of the solvent. Most of the reports are dealing with liquid propene and a polymerization period of 60 minutes
Thanks for your useful suggestion.
Before propylene polymerization: Propylene was introduced to the reactor kept at polymerization temperature after the toluene was saturated with propylene, MMAO was added and the solution was stirred for 10 min. In toluene, propylene is already in a saturated state.
In addition, our previous research results have shown that it is very easy to catalyze the polymerization of 1-olefins by phenyl substituted nickel α-diimine catalysts, so the P/D copolymerization can be successfully completed within 30 minutes at room temperature.
- Scheme 2: For R exponents should be used and no indices
Scheme 2. Chain-walking polymerization of propylene catalyzed with α-diimine nickel catalysts 1–5 [55,56].
- P. 3: The detailed description of the polymerization/copolymerization reactions can be summarized in order to avoid repetitions
Thanks for your useful suggestion. Propylene polymerization/copolymerization reactions have revised.
- P. 2: l.12 from the bottom: homo polymerization
Thanks a lot for catching this.
"home-polymerization" has been corrected to "homo-polymerization".
- P. 7, Table 3, top line: Branches has the exponent "d"
"c" has been corrected to "d".
- P. 8, l.5: enchainment
"1,8-enchainement " has been corrected to "1,8-enchainment".
The revised manuscript has been resubmitted to ‘polymers’. We look forward to your positive response.
Best wishes,
Fuzhou Wang
Institute of Physical Science and Information Technology,
Anhui University, Hefei 230601, Anhui, China.

Round 2
Reviewer 2 Report
The revised version of the manuscript considered most of my comments and I recommend publication. Two more remarks:
P.4, Scheme 2: The Rs in the Scheme should not have an index but an exponent P.9, l.8: no carbon atoms (instead of carbons)
Author Response
Title: “Living Chain-Walking (Co)Polymerization of Propylene and 1-Decene by Nickel α-Diimine Catalysts” Polymers 894363
Dear Reviewers,
Thanks for your useful comments. We have revised the manuscript accordingly, and detailed corrections are listed below point by point:
Reviewer: 2 The revised version of the manuscript considered most of my comments and I recommend publication. Two more remarks: P.4, Scheme 2: The Rs in the Scheme should not have an index. but an exponent P.9, l.8: no carbon atoms (instead of carbons)
Thanks for your useful suggestion.
The index has been removed in Scheme 2.
Scheme 2. Chain-walking polymerization of propylene catalyzed with α-diimine nickel catalysts 1–5.
"carbon atoms" has been corrected to "carbons".
The revised manuscript has been resubmitted to ‘polymers’. We look forward to your positive response.
Best wishes,
Fuzhou Wang
Institute of Physical Science and Information Technology,
Anhui University, Hefei 230601, Anhui, China.